# Environmental Benefits of Sleep Apnoea Detection in the Home Environment

Ragab Barika [1], Heather Elphick [2], Ningrong Lei [1], Hajar Razaghi [1] and Oliver Faust [1,*]

1   Department of Engineering and Mathematics, Sheffield Hallam University, Sheffield S1 1WB, UK
2   Sheffield Children's Hospital, NHS Foundation Trust, Sheffield S10 2TH, UK
*   Correspondence: oliver.faust@gmail.com

**Abstract:** Sleep Apnoea (SA) is a common chronic illness that affects nearly 1 billion people around the world, and the number of patients is rising. SA causes a wide range of psychological and physiological ailments that have detrimental effects on a patient's wellbeing. The high prevalence and negative health effects make SA a public health problem. Whilst the current gold standard diagnostic procedure, polysomnography (PSG), is reliable, it is resource-expensive and can have a negative impact on sleep quality, as well as the environment. With this study, we focus on the environmental impact that arises from resource utilisation during SA detection, and we propose remote monitoring (RM) as a potential solution that can improve the resource efficiency and reduce travel. By reusing infrastructure technology, such as mobile communication, cloud computing, and artificial intelligence (AI), RM establishes SA detection and diagnosis support services in the home environment. However, there are considerable barriers to a widespread adoption of this technology. To gain a better understanding of the available technology and its associated strength, as well as weaknesses, we reviewed scientific papers that used various strategies for RM-based SA detection. Our review focused on 113 studies that were conducted between 2018 and 2022 and that were listed in Google Scholar. We found that just over 50% of the proposed RM systems incorporated real time signal processing and around 20% of the studies did not report on this important aspect. From an environmental perspective, this is a significant shortcoming, because 30% of the studies were based on measurement devices that must travel whenever the internal buffer is full. The environmental impact of that travel might constitute an additional need for changing from offline to online SA detection in the home environment.

**Keywords:** sleep apnoea; artificial intelligence; polysomnography; remote monitoring; computer-aided diagnosis

## 1. Introduction

Healthy sleep is necessary for children and adults [1]. Sleep is a vital physiological activity that accounts for around one-third of a person's life [2]. A good night's sleep can help people to be more productive at work and have a more positive attitude in life [3]. Sleep deprivation can lead to cardiovascular disease (CVD), including stroke and coronary heart disease, endocrine problems, amnesia, and inattention, all of which have a negative impact on regular working and living conditions [4]. This was confirmed by systematic reviews and meta-analyses that linked these problems to shorter sleep durations [5]. A report from the World Congress of Cardiology and Cardiovascular Health in 2016 stated that CVDs are the leading cause of death worldwide, with an estimated total of >17 million fatalities [5]. The detrimental effects of sleep disorders on both mental and cardiovascular health make sleep disorder detection a public health priority.

Sleep apnoea (SA) is one of the most common sleep disorders that deteriorates mental and cardiovascular health. It affects nearly 1 billion people around the world [6], and it is difficult to be diagnosed [7,8]. At least 20% of all adults in developed countries suffer

from some form of SA [9]. The disorder is characterised by periods of shallow breathing (hypopneas) or no breathing at all (apnoea) [1,7,10]. In comparison to the general population, patients with untreated SA have a higher mortality rate [8]. The disease is associated with several comorbidities [10,11]. It has a variety of symptoms that might interfere with everyday activities [10], can lead to high blood pressure [12], CVD [8,11], type 2 diabetes mellitus (DM), and stroke [1,12,13]. The prevalence and severity of its symptoms make SA a public health problem. There are three main types of SA events, namely central sleep apnoea (CSA), obstructive sleep apnoea (OSA), and mixed sleep apnoea (MSA) [10]. The most frequent type of SA is OSA, which is caused by an obstruction of the airway during sleep [14–16]. CSA is caused by the brain failing to provide the proper signals to the muscles that govern breathing during sleep [17]. MSA (also known as complex sleep apnoea syndrome) is a combination of OSA and CSA [18]. It is estimated that 80–90% of SA cases are undiagnosed [19]. Therefore, cost-efficient, and sustainable diagnostic pathways are essential for addressing this public health problem. SA diagnosis is based on the Apnoea–Hypopnea Index (AHI), which measures the number of apnoea and/or hypopnea occurrences during one hour of sleep [20,21], as well as clinical criteria such as the daytime sleepiness caused by apnoea-related sleep disturbances. Recently, it has been argued that the AHI is vulnerable to clinical fluctuation and that an alternate metric to determine the OSA severity is needed. The morphology and length of apnoeas are not taken into consideration by the AHI, which is a significant disadvantage. Although the AHI system of severity grading is far from perfect, it has survived because of software that makes computing the AHI easier [22,23]. Overnight monitoring in a sleep lab is a common part of an evaluation. A polysomnography (PSG), often known as a sleep study, is a multi-part examination that measures and records particular physical actions while the patient is sleeping. A skilled sleep specialist analyses the recordings to determine if SA or other sleep disorders are present [24]. However, this approach is resource-intensive because the facility must be built and maintained. Furthermore, patients and sleep physicians must travel to the facility. This travel, together with the efforts of building and maintaining the facility, have a significant environmental impact. That means the current pathways for a SA diagnosis contribute to pollution, contamination, and destruction of the natural environment. Many attempts have been made in recent years to find an alternative device or approach that avoids the limitations of PSG [25]. Replacing central sleep labs with services based on a distributed infrastructure might reduce the environmental impact of SA diagnosis. These services could be established through remote monitoring (RM) technologies that incorporate mobile communication, cloud servers, and artificial intelligence (AI) [26,27]. Initially, SA detection and diagnosis support services based on RM technology were driven by cost pressure and patient comfort [28,29]. Widespread RM deployment and the associated SA diagnosis support systems will assist users in making appropriate and timely therapeutic decisions [30]. This will enhance the clinical outcomes [31], early and real-time SA detection, cost efficiency through fewer hospitalisations, and waiting list reductions [32,33]. It is expected that RM for SA detection will grow significantly in the next decade [34,35]. Given these wide-ranging changes, it is important to consider the environmental impact in RM-based SA detection.

SA detection in the home environment is an emerging technology. We hypothesise that adequate technology choices can lead to positive environmental impacts for the large-scale deployment of patient-led data acquisition. In this review paper, we argue that RM-based SA detection services have a lower environmental impact when compared with the standard sleep disorder detection methods. This advantage comes from reduced travel, for both patients and healthcare specialists, as well as resource sharing. To be specific, a shared communication and processing infrastructure allows us to establish SA detection services that can complement and, in some cases, replace sleep studies in the sleep lab. Having established the general benefits of SA detection services, we have turned our attention to specific systems that enable functionality of the service by reviewing 113 papers on that topic. The reviewed systems used a wide range of signals and methods for SA detection

in the home environment. As a result, these systems had varying levels of practicality. With respect to the environmental impact that arises from deploying specific RM-based SA detection systems, the most important property was whether the measurement evaluation was done online or offline. In general, offline systems require more effort to initiate and maintain the measurements. Moreover, offline measurement durations are limited by device-specific properties such as the available memory within the sensor. Online systems do not have this restriction. Furthermore, it is more difficult to establish resource sharing with offline systems compared to online systems. In this review, we established that just over 50% of the RM-based SA detection systems used online processing, and around 20% did not report that important property. That means at least 30% of the studies do not minimise their environmental impact. Another important finding of this review is the fact that environmental concerns did not feature in the reviewed articles. All the research work was driven by medical needs. Understanding, and indeed promoting, the environmental benefits of resource sharing and less travel through RM-based SA detection in the home environment might lead to more research funding being made available to create practical problem solutions. To the best of our knowledge, this is the first work that has established the environmental benefits of SA detection in the home environment.

The remainder of this manuscript is organised as follows. Section 2 provides some background on the methods used to detect SA in the home environment. Section 3 describes our article search methodology, while Sections 4 and 5 give our discussion and findings, respectively. Section 6 concludes the manuscript.

## 2. Background

Sleep is characterised by the suspension of consciousness or, during Rapid Eye Movement (REM) sleep, altered consciousness [36]. Unfortunately, there is no direct measurement of consciousness. This makes sleep disorder detection difficult. Therefore, a wide range of physiological signals is captured during a sleep study [37]. Usually, such a sleep study takes the form of a PSG. A PSG is recorded for at least one night, and the manual data analysis for each night can take up to 4 h. RM-based SA detection services are expected to acquire data over several nights—some systems have no restrictions on the amount of data they can acquire. Therefore, the manual analysis of data delivered by RM systems would be too demanding for human experts. Hence, an integral part for all SA detection services should be automated data analysis based on AI models. In the remainder of this section, we discuss the individual topics in detail.

### 2.1. Physiological Signals Used for Sleep Apnoea Detection

Physiological signals reveal how processes, within the human body, unfold over time. Such signals can provide objective evidence for transient disorders where symptoms are not always present. The American Academy of Sleep Medicine (AASM) has recommended the use of both a nasal cannula and a thermistor for the scoring of apnoeas and hypopneas since 2007 [38]. Hence, physiological signals are used for SA detection. As an alternative to the PSG, for the diagnosis of SA, signals can be observed on an oxygen saturation ($SpO_2$) recording alone if analysed by an experienced physician. The Heart Rate Variability (HRV) and an Electrocardiogram (ECG) can also indicate a suspicion of SA.

#### 2.1.1. Electrocardiogram (ECG)

An ECG is used to diagnose a variety of cardiovascular disorders, including coronary heart disease and cardiac arrhythmias. ECG signals are recordings of the electrical activity of the human heart over time [16]. Several research studies have found that ECGs from different people have some similarities, indicating that using only ECG sensors can achieve a good SA detection accuracy. However, due to the low sensitivity and specificity, this measure is not used alone in clinical practice but is observed alongside measurements such as respiratory airflow and $SpO_2$ [9,39]. In the absence of heart diseases, ECG signals are highly structured, and individual signal components can be identified through visual

inspection. The ECG trace is made up of several waves that are labelled P, QRS, and T. Each wave corresponds to a different physiological event during the cardiac cycle [7]. The breathing rate is linked to the heart rhythm via the autonomous nervous system [40]. It was observed that, when breathing stops, the heart rhythm slows down [41]. As the time with no breathing increases, the subject becomes tense, and the heart rhythm speeds up again. Morphological variations in the ECG signals reflect these changes. Hence, these signals can be used as an objective measure to detect SA [42].

### 2.1.2. Heart Rate (HR)

A HR signal is composed from consecutive beat-to-beat intervals of the human heart [43,44]. As such, the HR is the most widely measured physiological signal [45,46]. The beat-to-beat intervals are usually extracted from either an ECG or photoplethysmogram (PPG) signals [47]. HRV is a physiological parameter that measures the variations in the time interval between consecutive heartbeats in milliseconds. It is regularly measured to provide objective evidence that supports a CVD diagnosis, since it is linked to heart health. High HRV values are often connected with a healthy cardiac condition, and so, a lower death probability can be established. SA episodes change the heart rhythm, and these changes will be reported by the HRV directly. It is possible to detect these changes and thereby establish an objective measure for SA. However, gender and age of the patient may have an impact on the HRV [48]. An important environmental benefit of HR measurements arises from the fact that the human heart beats around once every second. The resulting data rate is approximately one sample a second. The very low data rate makes communication resource reuse straightforward. Furthermore, the low data rate implies that the energy requirement for a signal analysis is also low.

### 2.1.3. Oxygen Saturation of the Blood ($SpO_2$)

Single biological markers, like $SpO_2$, have been employed in several studies to detect SA [49,50]. The AASM Task Force has included blood oxygen saturation as one of the measurements that characterises SA and hypopnea episodes [50]. The amount of oxygen that is saturated in haemoglobin is referred to as $SpO_2$ [43]. A healthy person's oxygen saturation level is usually between 95 and 100% [51]. Oxygen levels of 90–95% are still considered safe for healthy subjects, but dangerous for patients with chronic lung diseases. $SpO_2$ values can be categorised as flows: normal and healthy arterial level ($SpO_2$ within 95–100%), mild hypoxemia ($SpO_2$ within 91–94%), hypoxic (arterial level of $SpO_2$ is within 85–94%), and severely hypoxic (arterial level of $SpO_2$ below 85%). It is reported that oxygen levels below 90% are dangerous and that oxygen levels below 80% are harmful to vital organs [43,52]. Most studies use $SpO_2$ and ECG signals because of their link to apnoeic events. Research has shown that the HR and systolic blood pressure rise in response to apnoeic episodes [53]. Burgos et al. used $SpO_2$ measurements to detect SA [50].

### 2.1.4. Polysomnography (PSG)

SA is generally diagnosed and treated in sleep laboratories using PSG, which is associated with significant waiting times for patients and high costs [54,55]. To conduct a sleep study, patients must spend at least one night in the sleep lab with several electrodes attached to their body [56]. These electrodes might disturb a patient's sleep, resulting in measurement data variations [37]. Since it must be performed in a sleep lab with physicians, the diagnosis results may be influenced by the lab environment, as well as the intrusive and inconvenient measurement sensors attached to the patient's body [9]. Many patients have trouble sleeping in such an environment. Due to the presence of numerous leads and monitors, some patients report feeling constrained during in-laboratory PSGs, resulting in them spending more time in the supine position than they would during a typical night at home [7,57,58]. A PSG requires gathering 12 separate signals with a minimum of 22 lead wires linked to the patient's body, making a signal analysis difficult and causing discomfort to the patient [38]. Intrusiveness and restricted availability make PSGs unsuitable for

screening purposes [51]. There is a lack of facilities and a lack of sleep specialists, resulting in extremely long waiting periods for patients [59]. Furthermore, manually analysing and scoring sleep using PSG traces is a time-consuming task [52]. It can take 2–4 h to score all data acquired during a full night's sleep [5].

*2.2. Automated Apnoea Detection*

AI models can extract objective information from physiological measurements for automated SA detection. These decision support models become essential in long-term monitoring because a manual analysis is impractical for the acquired data volume. For example, an advanced RM-based SA detection service might acquire ECG signals while the patient is sleeping. These signals are communicated in real time to a central cloud server, where they are available for analysis. Such a measurement setup poses no restriction on the amount of data collected, i.e., it is possible to record the ECG every night. A manual analysis would demand that a sleep physician read 6–8 h of ECG every day to monitor one patient. Furthermore, SA detection services in the home environment are scalable, and therefore, a manual analysis would be required for several patients. Hence, automated decision support is essential for any meaningful SA detection service in a home environment. To address that need, scientists created a wide range of AI based SA detection models. These models were based on technologies such as: ECG-Derived Respiration (EDR) [60], Classification and Regression Tree (CART) [61], Statistical Classifier (SC) [62], Convolutional Neural Network (CNN) [63–66], Recurrent Neural Network (RNN) [67], K-nearest Neighbour (KNN) [68], and Support Vector Machine (SVM) [69].

Our background research shows that the tools and techniques are available to establish RM services for SA detection. In the next section, we review systems that establish these services in the home environment. With respect to the discussion of these systems, we are especially interested in the properties that allow us to determine their environmental impact.

## 3. Sleep Apnoea Detection in the Home Environment

In this section, we outline our approach to review SA detection systems for the home environment. We conducted a comprehensive search across Google Scholar to find all research articles on the topic of automated SA detection in the home environment that were published between 2018 and 2022. We chose this period, because there was a lot of forward-thinking work on AI during that time. The database was queried using predefined Boolean search terms. Table 1 shows that the single search term "apnea home" returned 179 results. These articles were filtered according to the Preferred Reporting Items for Systematic Reviews and Meta-Analyses (PRISMA) technique [70]. Figure 1 shows the PRISMA flow diagram, which documents the article filtering and refinement process. During the filtering, we eliminated duplicate entries, review articles, conference papers, non-English publications, and manuscripts without ACC results. Overall, the filtering process eliminated 65 papers, and we were left with 113 original research publications.

**Table 1.** Boolean search strings.

| Title | AND (Full-Text and Metadata) | Database | No. of Studies |
|---|---|---|---|
| "Apnea home" | "Apnea home" | Google Scholar | 179 |

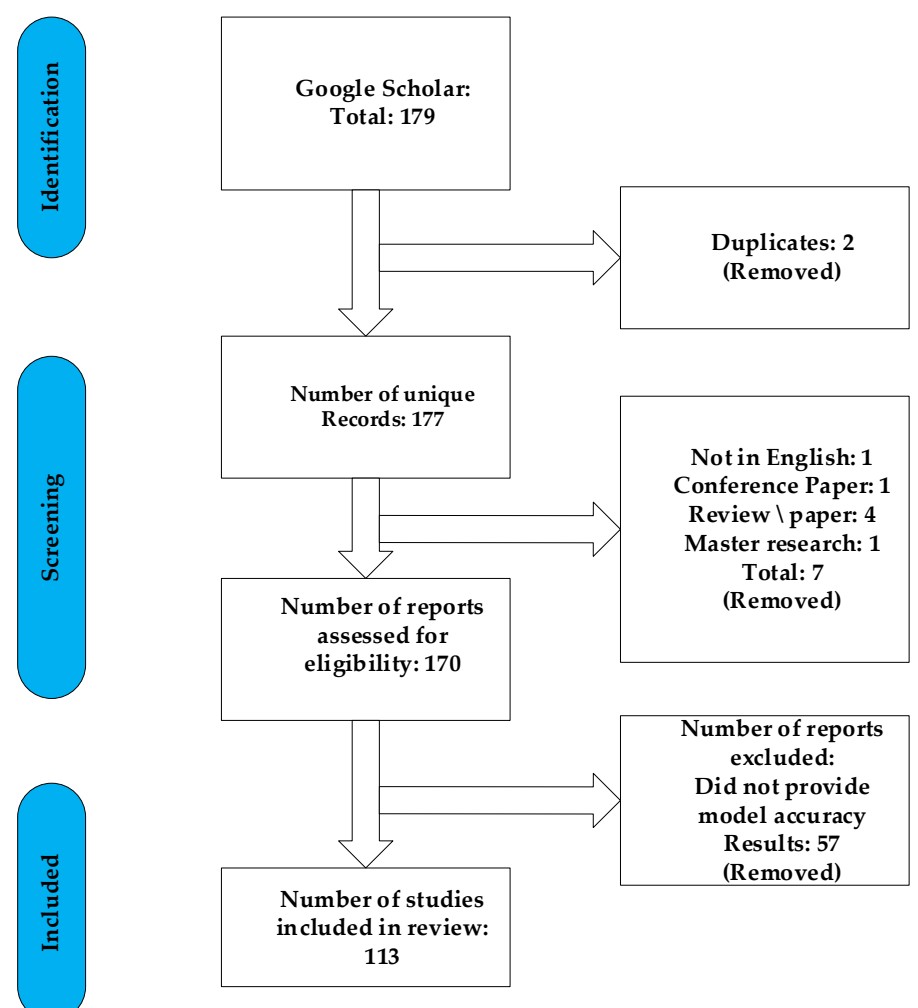

**Figure 1.** Flow chart of the PRISMA model for article selection.

## 4. Results

The 113 articles on SA detection systems in the home environment were analysed in terms of the signal used, detection method, data handling, number of participants, and detection performances. The signals used for SA detection have a significant impact on the environmental footprint. Different physiological signals have different requirements in terms of the measurement setup, communication bandwidth, storage capacity, and processing capabilities. As such, knowing the signal will indicate the resources needed to establish an SA detection service for the home environment. Evaluating the detection method together with the number of participants allows us to reason out the technology readiness level. Table A1 in Appendix A provides an analysis result summary.

Table A1 details the detection method used and the SA detection performance for each of the 113 studies under review. The best detection performance, in terms of ACC score, was reached by a few of the investigations. However, these results were based on "apnea home" investigations, hence the findings may not reflect a widespread trend. We excluded duplicate items, review articles, non-English publications, Master's dissertations, and works irrelevant to our criteria. Furthermore, during this investigation, we discovered other papers that were relevant to our criteria, although these articles only had an abstract. In this case, we had to remove these articles from our work. As seen in Table A1, several signals have been employed in a range of different studies. Each signal has different purposes. The signals that were used in this article are PSG, SpO2, home respiratory polygraphy (HRP), home polygraphy (HPG), ECG, seismocardiography (SCG), PPG, polygraphy (PG), respiratory inductance plethysmography (RIP), audio, and HR. Figure 2 shows the signals

used by the 113 evaluated studies. Table A1 in Appendix A provides further details on these SA detection studies. Figure 3 shows the approaches used in this work to detect SA. The numbers in the pie charts represent the amount of research articles that reported each method. The pie chart in Figure 2 reveals that PSG signals are the most-studied method among the examined publications, with 78 total research articles using this method. There were eight studies using HR signals and three studies using ECG. SpO2 was only utilised in one study. Twenty-three research articles employed other signals, as shown in the pie chart in Figure 3. Machine learning and deep learning were utilised four times each, as seen in Figure 3. Sleep physicians were the most reported to detect SA in a total of 30 studies; 75 studies used other methods. Figure 4 depicts the data management approaches, which included 35 online studies, 30 offline research, and 48 unreported studies. Figure 5; Figure 6 show the number of participants and the accuracy distribution. It can be observed that 101 people participated in this study, with 12 not participating. Furthermore, Figure 6 depicts the specifics of 60 studies that reported their accuracy and 53 studies that did not report their accuracy.

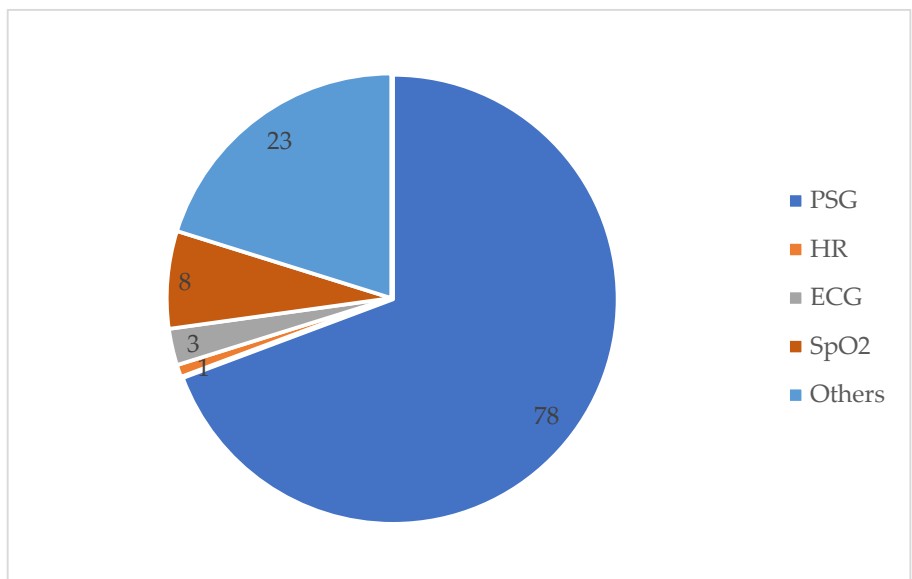

**Figure 2.** PGS, HR, ECG, SpO2, and others are the signals used to detect SA.

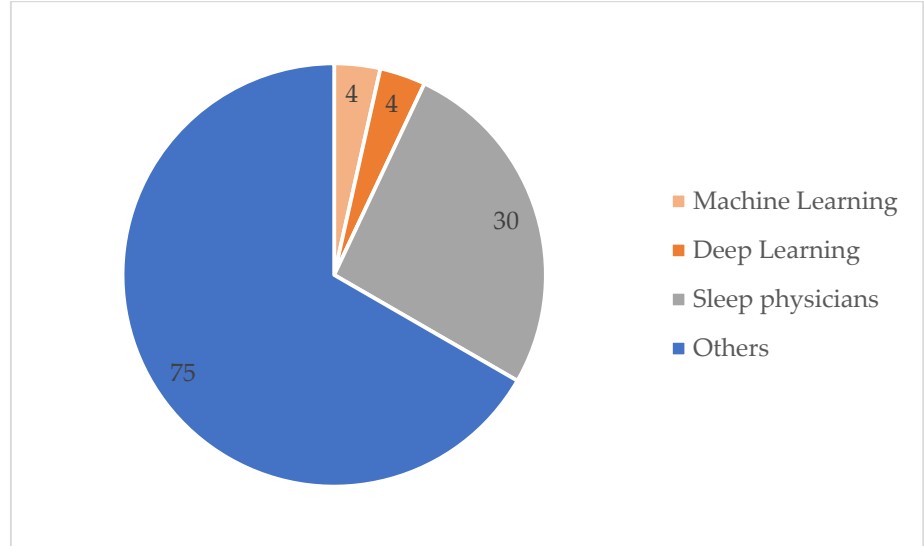

**Figure 3.** SA detection method.

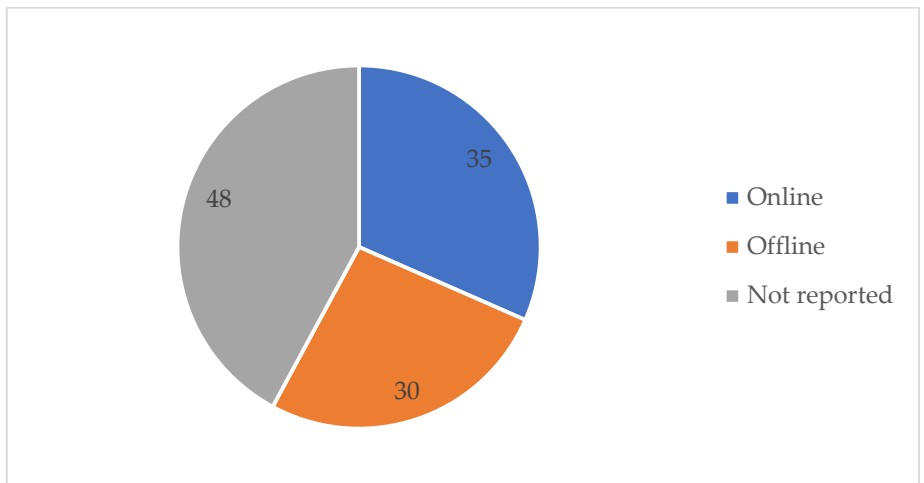

**Figure 4.** Data handling method.

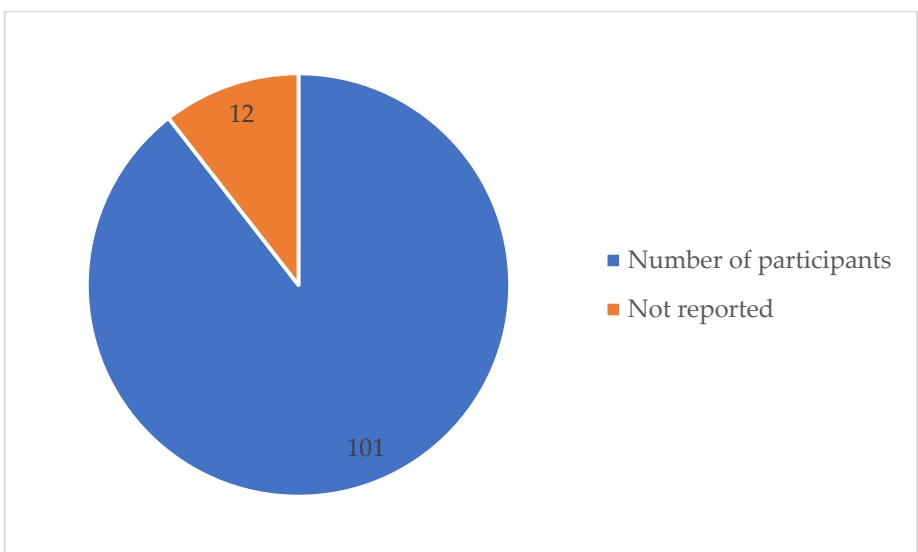

**Figure 5.** Number of participants reported.

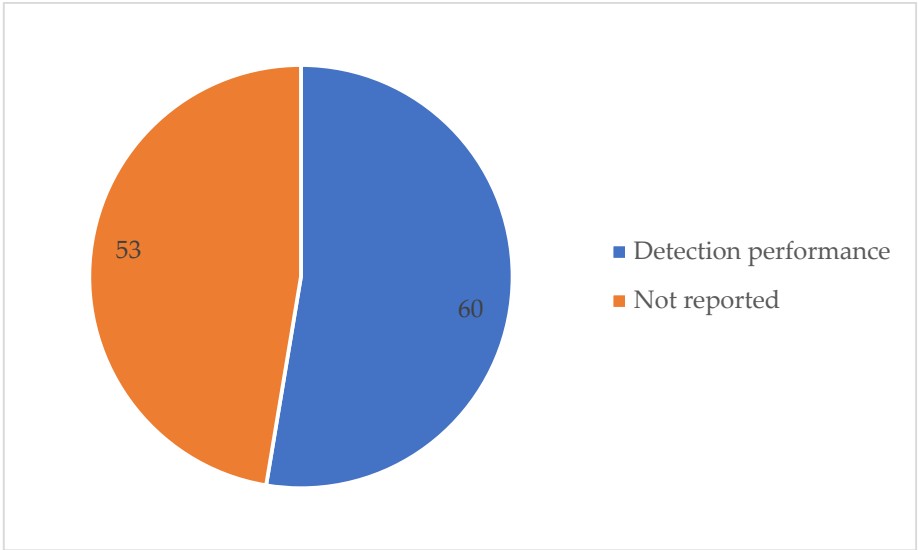

**Figure 6.** SA detection performance stated.

## 5. Discussion

The environmental impact of specific actions has become a major societal concern, affecting both business and organisational competitiveness. This has sparked an interest in objective research on this topic, which is based on collecting and analysing data. The term "environmental impact" refers to a change in a situation's outcome. The result of an action or event that affects the social, environmental, or economic well-being is referred to as an outcome [71]. The environmental impact of an activity is significant, and it demands immediate attention to mitigate the negative consequences on global health. The reduced use of equipment may be the most appropriate and effective strategy to minimise an environmental impact. Healthcare is a resource intensive activity and therefore it is important to discuss the environmental impact.

In this study we focused on the environmental impact of SA detection. It has been well established that SA is a significant health and economic issue, particularly in developed countries [72]. As such, SA is the most prevalent sleep condition for which diagnostic testing is performed at sleep labs. This testing takes the form of a PSG, which requires a comprehensive monitoring system to record a range of physiological signals during sleep. The PSG results could take a few weeks to arrive [73]. This type of investigation is very resource-intensive because a certified sleep technologist must set up and monitor the data acquisition equipment for the duration of the measurement, which is usually a whole night. Subsequently, the data must be analysed by a sleep physician, which takes up to four hours. This results in a resource shortage, which causes longer travel distances for patients [74]. From an environmental perspective, this travel is a significant drawback of the PSG. Another detrimental effect on the environment comes from the fact that a dedicated sleep lab needs to be built and subsequently maintained.

RM may be the most effective strategy to limit travel outside of the home environment and lower the risk of diseases like SA. Apart from reducing travel, RM-based SA detection services for the home environment can benefit healthcare providers through automating the SA detection process. The possibility of reducing healthcare costs is one of the most compelling reasons for introducing RM, followed by the desire to improve healthcare access. The savings from RM are not consistently reported in the research literature. Jiménez-Marrero et al. [75] put forward that RM-based SA detection offers significant cost savings. In contrast, Lew et al. [76] found only minor cost savings when they studies admission expenses. To be specific, for some patient categories, the cost only decreased from USD 10,835 to USD 10,678. These findings led to the conclusion that cost savings were a modest EUR 188 per person per year [77]. Other studies found that RM-based SA detection services have the same or even higher costs [29]. Many economic assessments of RM reflect only direct healthcare costs and do not include the overall programme costs like equipment amortisation or service costs. Other factors may also have an impact on the results. For example, RM-based hypertension and congestive heart failure detection is less expensive than the distant monitoring of respiratory illnesses [29].

In recent times, RM and inconspicuous sensors, cloud computing, and enhanced internet connectivity have all improved the technology for monitoring, aiding, and enhancing human health. For instance, the Internet of Things (IoT) paradigm's presence and quick growth has had an impact on how individuals track their health [78]. Moreover, most of today's wearable devices can track the HR and physical activity. More appliances are equipped with an internet connection, and RM is becoming more prevalent. Unobtrusive sensor data can provide a more thorough picture of the health and lifestyle habits of care recipients. As a result, technology has a direct impact on the ability of elderly and disabled individuals to detect their health at home and live more independent lives [79]. From the environmental perspective, the pervasive use of RM technology is a step in the right direction, because it allows us to administer more care with the same or marginally increased resource requirements [80]. A SA detection service in the home environment might become one of many servicers offered by a healthcare platform. The core modules of such a platform

facilitate data acquisition, communication, and storage. Specific modules will customise the platform to offer unique services, such as SA or atrial fibrillation detection [26].

During our review, we looked at the AI models that can be used for a computer-aided SA diagnosis. In a second phase, we looked at RM systems for SA detection in the home environment. For AI models, the overarching trend was that the research output focused on automated SA detection is growing, as shown in Figure 1. That means it is an active field, and we can expect a continuous improvement of the materials and methods for AI-based SA detection. Coupled with the RM techniques, this is certainly beneficial for the environment. However, we came across another trend when reviewing AI-based SA detection, namely the emergence of deep learning. The environmental impact of this trend is not clear. The computational complexity of deep learning models is significantly higher when compared to classical machine learning [81]. Therefore, it takes more energy to design the model. However, the trade-off here is to replace manual design work, in the form of feature engineering for classical machine learning, with automated feature extraction, which happens when we train deep learning algorithms. It seems unlikely that the increased energy requirement for designing deep learning models will be a serious barrier for this technology, especially when we consider the alternative being human labour that is focused on feature engineering.

The review on SA detection in the home environment revealed that most systems rely on PSG measurements to gain objective information. From an environmental perspective, this is not ideal, because PSG signal measurements require a complex measurement setup, which is resource intensive. For example, it might be necessary that a sleep technician or a nurse travel to the home environment of the patient to establish the measurement setup. Compared to the measurement setup of a PSG, individual signals, such as HR, ECG, and SpO2, are more straightforward. HR signal acquisition requires the least measurement setup, which might enable patient-led data acquisition. For example, a patient attaches the sensor and ensures that the data is relayed to a cloud server that runs a deep learning model for automated SA detection. Such a service would have very little environmental impact because both communication infrastructure and cloud server facilities are shared, which causes minimal additional energy expenditure. During service deployment, only the sensor constitutes additional hardware that needs to be produced and maintained. Even the sensor hardware could be shared amongst multiple services. This sensor sharing idea is based on the fact that HRV is a good predictor of human health. Hence, HR measurements can be used to detect and monitor a wide range of diseases, including, but not limited to, heart arrythmias, diabetes, and epilepsy [70]. The resource sharing could be facilitated by a healthcare platform that offers SA detection as one of many services. From an environmental perspective, both infrastructure and sensor reuse are strong arguments that the benefits of the platform approach outweigh the additional burden on the environment.

According to Rosenberg et al. [82], the utilisation of technology, such as RM, in the home environment for the support and care of people with SA is crucial. They reflect the costs that burden citizens with SA, as well as the anxiety that comes with the long waiting times for sleep lab-based diagnoses. According to the research, RM is the most desirable feature for SA detection in the home environment. The implementation of such technology has the potential to improve patient care while also reducing the demand for both resources and medical services. This might reduce the financial strain on healthcare systems. Moreover, the environmental benefits of using RM in SA detection are becoming more widely recognised. It is an unattended instrument that does not require the presence of a laboratory attendant. Individuals can use the monitor at home if they follow the technician's recommendations. Although the sensitivity of SA detection services in the home environment is currently lower than that of PSG, it saves time and money for patients while also providing convenience and comfort.

Looking beyond the current economic costs and system capability reveals future trends that might offer opportunities for businesses and healthcare providers. Smartphones and tablets are equipped with an increasing number of sensors that collect a large amount of

personal data in various formats and for various purposes [83]. For a manual analysis, this constitutes a problem because of the limited availability of sufficiently qualified human labour. However, technology improvements are not limited to the communication infrastructure; it is also projected that AI models for data analysis will continue to improve. Understanding of the algorithms and the availability of a cost-efficient parallel processing infrastructure are the two main drivers for that progress. Hence, as our ability grows to measure physiological signals in the home environment, the progress in AI technology will ensure that the data can be utilised. One use of such data is SA diagnosis support. Technology improvements are general trends that lead to gradual changes. As a result, we predict that there will be a broad acceptance of using more and more data-driven healthcare. It is likely that ethical issues are addressable with technological solutions, such as data encryption to address privacy concerns. Considering the environmental impact of actions like establishing a service platform for patient RM might open an independent line of argument to justify future decisions. To be specific, the environmental impact should be considered alongside ethical concerns, technological feasibility, and economic costs.

*5.1. Limitations*

There are certain limitations to this paper. First, it is possible that the literature search missed some important papers. Second, not all facets of SA disorders were covered. Third, several of the topics covered lacked high-quality data. Fourth, there was a lack of low cost and readily available RM-based SA detection services that could be used in the home environment. Fifth, patient RM requires internet access, which may not be available in some areas.

During the review, we learned that the environmental impact is hard to quantify, because there are a vast number of factors, even when environmental pollution is considered on its own. Therefore, the best support we have for promoting RM-based SA detection services is that increasing the use of this technology would benefit public health with a moderate environmental impact. Some readers might be dissatisfied with this statement because it appears vague. The statement becomes more concrete when we consider the alternative, which would be to build and maintain more sleep labs. Clearly, more sleep labs would have detrimental effects on the environment.

*5.2. Future Work*

SA is a life-threatening condition that affects people all around the world. The rapid rise in the number of SA sufferers each year is putting governments under a lot of financial strain. Several SA treatments have been proposed to alleviate or cure the condition. However, there is a scarcity of research comparing these treatments. As a result, a comprehensive guideline for selecting an appropriate treatment for people with various degrees of SA is required. Future research should incorporate the following to create a thorough evidence-based comparison to advise patients and doctors:

With the expanding use of RM around the world and the growing number of people who use it, RM is becoming increasingly important in terms of enhancing patient care, safety, and comfort. For the patients and the healthcare team, RM is an essential technology based on shared resources. Resource sharing and computational decision support result in the fact that upscaling the technology use has a low environmental impact. Therefore, in the future, we should see widespread deployment of this technology.

To reduce sensor waste, it is critical for companies to develop alternative instruments that can be used by patients at home and that can be useful for reducing the workload of both medical staff and patients coping with disease to reduce environmental waste. RM is one tool that can help in this situation.

To enhance healthcare, there is a focus on individuals who suffer from chronic diseases such as SA. Improved decision support algorithms and an appropriate healthcare network can aid the patient with their illness, and the algorithms could also help the doctor to predict, diagnose, and treat a problem. Algorithms could explain and anticipate how

patients interact with their healthcare providers to make health decisions. In addition, algorithms are critical for detecting risk changes on a continuous basis. AI models could choose a sequence of self-performed actions for the patient to manage that risk, such as to increase physical activity or improve the adherence to a prescribed medication regime [84].

## 6. Conclusions

SA is a major health and economic problem, especially in developed countries. Therefore, physical problem solutions that address or indeed attempt to address SA detection have an impact on the environment. To establish the environmental benefits of RM-based SA detection, we studied the enabling technologies, and we reviewed systems that detect SA in the home environment. During these activities, we learned that physiological signals and their analysis play a central role in SA detection. The most dynamic enabling technology is AI-based SA detection. We found that the research in this field is expanding with the emergence of novel deep learning approaches. However, this continued interest and, indeed, the associated research outputs have not percolated through to practical systems for SA detection in the home environment. Only 8 out of 113 studies used AI techniques for SA detection. Hence, there is room for improvement, especially when we consider the second important review finding, namely the apparent lack of online decision support.

SA detection and diagnosis support services based on RM technology constitute progress. In this paper, we argue that this progress can be achieved without a significant environmental impact. To be specific, these services can be established by reusing existing infrastructure. However, we also recognise that sleep labs will continue to play a vital role in the future for diagnosing sleep disorders that are not yet detectable through remote monitoring and for research purposes. Hence, RM will allow us to diagnose more SA earlier, and this will improve the outcomes for patients with the same or marginally more resources. SA detection and diagnosis support services can be established by reusing the available infrastructure. From an environmental perspective, the infrastructure is already built, and there is no, or at least a significantly reduced, need to construct new dedicated sleep labs. Another important advantage is the geographical and temporal decoupling of patient and physician. This decoupling is not only convenient for all parties involved, but it also reduces the administrative efforts required to synchronise and manage patients and healthcare professionals. Geographical decoupling leads to less mandatory traveling, which is an environmental advantage of RM-based SA detection and diagnosis support services when compared with traditional sleep studies.

**Author Contributions:** R.B.: writing—original draft preparation. H.E., N.L., H.R. and O.F.: all had equal roles as key contributors, discussing the fundamental concepts and the outline, providing critical comments on each part, and assisting in the shaping and composition of the work. All authors have read and agreed to the published version of the manuscript.

**Funding:** This research received no external funding.

**Acknowledgments:** The authors would like to express their gratitude to Sheffield Hallam University's Electrical and Electronic Department for their continuous cooperation and support.

**Conflicts of Interest:** The authors declare no conflict of interest.

## Abbreviations

The following abbreviations are used in this manuscript:

| | |
|---|---|
| PRISMA | Preferred Reporting Items for Systematic Reviews and Meta-Analyses |
| AASM | American Academy of Sleep Medicine |
| AHI | Apnoea-hypopnea index |
| AI | Artificial Intelligence |
| CART | Classification and Regression Tree |
| CNN | Convolutional Neural Network |
| CSA | Central sleep apnoea |

| CVD | Cardiovascular Disease |
| --- | --- |
| ECG | Electrocardiogram |
| EDR | ECG derived respiration |
| GDP | Gross Domestic Product |
| HPG | Home polygraphy |
| HR | Heart Rate |
| HRP | Home Respiratory Polygraphy |
| HRV | Heart Rate Variability |
| IoT | Internet of Things |
| IT | Information Technology |
| KNN | K-Nearest Neighbour |
| MSA | Mixed sleep apnoea |
| OSA | Obstructive sleep apnoea |
| PG | Polygraphy |
| PPG | Photoplethysmogram |
| PSG | Polysomnography |
| RIP | Respiratory inductance plethysmography |
| RM | Remote Monitoring |
| RNN | Recurrent Neural Network |
| SA | Sleep Apnoea |
| SC | Statistical Classifier |
| SCG | Seismocardiography |
| SVM | Support Vector Machine |

## Appendix A

**Table A1.** Details of the 113 selected studies on SA detection in the home environment.

| Authors | Signal | Detection Method | Online/Offline | Number of Participants | Detection Performance |
| --- | --- | --- | --- | --- | --- |
| Saletu et al., 2018 [85] | PSG | Sleep physicians | Online | 265 | - |
| Massie et al., 2018 [86] | PSG | Sleep physicians | Online | 101 | - |
| Rosen et al., 2018 [87] | - | Home sleep apnoea test | Online | - | - |
| Ng et al., 2019 [88] | PSG | Sleep physicians | | 316 | Sensitivity = 78% Specificity = 23% Negative predictive value = 67% positive = 35% |
| Gu et al., 2020 [89] | SpO$_2$ | Sleep physicians | Online | 50 | Sensitivity = 85% Specificity = 87%% Positive and negative predictive value = 0.88% and 0.83% |
| Chiner et al., 2020 [90] | Home respiratory polygraphy HRP | Sleep physicians | Online | 121 | Accuracy = 93% |
| Gutiérrez-Tobal et al., 2019 [91] | SpO$_2$ | Machine learning AB-LDA | Offline | 230 | Accuracy = 78.7% |
| Zancanella et al., 2022 [92] | PSG | EmblettaX100 system | Offline | 40 | - |
| Manoni et al., 2020 [93] | PSG | MORFEA | Online | - | - |

**Table A1.** *Cont.*

| Authors | Signal | Detection Method | Online/Offline | Number of Participants | Detection Performance |
|---|---|---|---|---|---|
| Kole 2020 [94] | | Home sleep apnoea testing | - | >800 | - |
| R. Stretch et al., 2019 [95] | PSG | Sleep physicians | Online | 613 | Sensitivity = 0.46, Specificity = 0.95% Positive predictive value = 0.81% negative predictive value = 0.80% |
| Castillo-Escario et al., 2019b [96] | PSG | MATLAB | Offline | 13 | Sensitivity = 76%, Positive Predictive Value = 82% |
| Hunasikatti 2019 [97] | PSG | Sleep physicians | Online | 206 | - |
| Romero et al., 2022 [98] | PSG | Sleep physicians | Online | 103 | Sensitivity = 79% Specificity = 80% |
| Massie, Van Pee, & Bergmann 2022 [99] | PSG | WatchPAT | Offline | 20 | - |
| Kristiansen, Nikolaidis, et al., 2021 [12] | PSG | Machine learning | Online | 579 | Accuracy = 89% |
| Nobuaki Tanaka et al., 2021 [100] | - | W-PAT | - | 776 | - |
| Colelli et al., 2021a [101] | HSAT | Sleep physicians | Online | 119 | - |
| Ikizoglu et al., 2019 [102] | PSG and HPG | Sleep physicians | Online | 19 | Sensitivity = 100% Specificity = 83% |
| Aielo et al., 2019 [103] | PG | Sleep physicians | Online | 300 | Accuracy = 95% |
| Zavanelli et al., 2021 [104] | ECG, SCG, and PPG | Sleep physicians | Online | - | Accuracy = 95% |
| Colaco et al., 2018 [105] | PSG | Sleep physicians | Online | 43,780 | - |
| Ekiz et al., [106] | PSG | Sleep physicians | Online | 43,780 | - |
| Maggio et al., 2021 [107] | PSG | Embla® Embletta® GOLD portable sleep system | Online | 45 | Accuracy = 93% |
| Steffen et al., 2021 [108] | PSG and HST | Sleep physicians | Online | 131 | - |
| Orr et al., 2018 [109] | PSG and HST | MATLAB | Offline | 27 | Sensitivity = 70% Specificity = 71% |
| Gutiérrez-Tobal et al., 2021 [110] | SpO$_2$ | Least-squares boosting algorithm | Offline | 8762 | Accuracy = 87.2% |
| Fietze et al., 2022 [54] | polygraphy (PG) | Sleep physicians | Online | 505 | - |

**Table A1.** *Cont.*

| Authors | Signal | Detection Method | Online/Offline | Number of Participants | Detection Performance |
|---|---|---|---|---|---|
| Fitzpatrick et al., 2020 [111] | PSG | BresoDx® portable monitor | Offline | 233 | Sensitivity = 85% Specificity = 0.48% Positive and negative predictive values were, 0.81% and 0.54% |
| Ferrer-Lluis et al., 2019 [112] | Pulse oximetry | Apnealink™ Air | Offline | - | - |
| Huysmans et al., 2021 [113] | PSG | Total Sleep Time (TST) | Offline | 183 | Sensitivity = 78% Specificity = 89% |
| Joymangul et al., 2020 [114] | Positive Airway Pressure (PAP) therapy | Python | Online | 668 | - |
| Młyńczak et al., 2020 [115] | PSG | Audio sensor | Online | 30 | Accuracy = 86% Sensitivity = 96%, Specificity = 76% |
| Van Pee et al., 2022 [116] | PSG and PAT HSAT | Sleep physicians | Online | 167 | - |
| Castillo-Escario et al., 2019a [117] | audio signals | MATLAB | Offline | 3 | Accuracy = 95.9% |
| Navarro-Martínez et al., 2021 [118] | pulse oximetry | Epworth sleepiness scale, STOP-BANG questionnaire, and C-reactive protein screening | Online | 117 | Sensitivity = 80% Specificity = 92% |
| Patel et al., 2018 [119] | PSG | ApneaLink Air devices | Online | 106 | Sensitivity = 82% Specificity = 92% |
| Magalang et al., 2019 [120] | Nasal pressure | Fifteen HSAT | Offline | - | - |
| Muñoz-Ferrer et al., 2020 [121] | PSG | Sleepwise (SW) | Online | 38 | Accuracy = 84% |
| Light et al., 2018 [122] | EEG and PSG | Sleep physicians | Online | 207 | Accuracy = 95% |
| Oceja et al., 2021 [123] | PSG | HRP | Online | 320 | - |
| Di Pumpo et al., 2021 [124] | - | WatchPAT | - | - | - |
| Hoshide et al., [125] | PSG | CPAP therapy | Online | 105 | Accuracy = 86.9% |
| Hui et al., 2018 [126] | PSG | Respiratory polygraphy | Online | - | Accuracy = 95% |

**Table A1.** *Cont.*

| Authors | Signal | Detection Method | Online/Offline | Number of Participants | Detection Performance |
|---|---|---|---|---|---|
| Goldstein et al., 2018 [127] | PSG | Sleep physicians | | 196 | Accuracy = 84% |
| Jensen et al., 2022 [128] | PSG | NightOwl™ | Offline | 150 | Accuracy = 95% |
| Simonds 2022 [129] | Body movement, respiratory rate, heart rate, snoring, and breathing pauses | Withings Sleep Analyzer | Online | 67,278 | Sensitivity = 88% Specificity = 88% |
| Rajhbeharrysingh et al., 2019 [130] | PSG | Machine learning | Online | 14 | Accuracy = 82.9% Sensitivity = 88.9%, Specificity = 76.5% |
| Facco et al., 2019 [131] | PSG | Sleep physicians | Online | 43 | 80.0% |
| Kristiansen et al., 2021 [132] | PSG and PG | Sleep physicians | Online | 34 | Sensitivity = 97.2% Positive prediction value = 94.2%. |
| Li et al., 2021 [133] | PSG | Sleep physicians | Online | 43,780 | - |
| Massie et al., 2022 [134] | PSG | MATLAB | Offline | 261 | Sensitivity = 87% Specificity = 89% |
| Hart et al., 2021 [135] | PSG | CPAP | Offline | 18 | - |
| da Rosa et al., 2021 [136] | PSG | Sleep physicians | Online | 94 | Accuracy = 80.7% |
| Ashley Suniega et al., 2019 [137] | PSG | HRP | Online | 430 | Accuracy = 95% |
| Mosquera-Lopez et al., 2018 [138] | PSG | Machine learning | Offline | 14 | Accuracy = 86.96% Sensitivity = 81.82% Specificity = 91.67%. |
| Lipatov et al., 2019 [139] | PSG | HSAT devices | Offline | 141 | - |
| Silva et al., 2021 [140] | PSG | SPSS software | Offline | 427 | - |
| Bonnesen et al., 2018 [141] | Audio | Portable device | Online | 23 | Sensitivity = 75%, Accuracy = 60% |
| Green et al., 2022 [142] | PSG | Online video technician | Online | 100 | - |
| Ben Azouz et al., 2018 [143] | PSG | Equivital™ EQ02 LifeMonitor | Online | 32 | - |
| Honda et al., 2022 [144] | Respiration activity | wearable sensor | Offline | - | - |
| Ghandeharioun 2021 [145] | ECG and SpO$_2$ | Sleep physicians | Online | 155 | Accuracy = 85% |

**Table A1.** *Cont.*

| Authors | Signal | Detection Method | Online/Offline | Number of Participants | Detection Performance |
|---|---|---|---|---|---|
| Labarca et al., 2018 [146] | PG | HSAT an Embletta® | Online | 198 | - |
| Lee et al., 2021 [147] | PSG | HSAT | Offline | 154 | Sensitivity = 85% Specificity = 95% |
| Huysmans et al., 2020 [148] | ECG and RIP | CNN | Online | 81 | Kappa score = 0.48 |
| Barriuso et al., 2020 [149] | Respiratory polygraphy | HRP | Online | 301 | - |
| Mashaqi et al., 2018 [150] | PSG | HSAT, RYGB and LSG | Online | 10 | Accuracy = 94% |
| Takao et al., 2019 [151] | Audio | Autoencoder | Offline | 5 | Accuracy = 94.7% |
| Borsini et al., 2021 [152] | PG | Apnea Link Plus and Air | Online | 3854 | Accuracy = 90% |
| Gu, W., & Leung 2018 [153] | PPG | pulse oximeter | Online | 23 | Accuracy = 97% |
| Mieno et al., 2020 [154] | PSG | PulSleep LS-140 | Offline | 58 | Sensitivity = 96.4% Specificity = 100% |
| Arguelles et al., 2019 [155] | PSG | HSAT | Online | 88 | Accuracy = 98% |
| Stretch et al., 2019 [156] | PSG | k-nearest neighbors algorithm | Offline | 415 | Sensitivity = 0.43% Specificity = 0.96% |
| Iqubal & Lam 2020 [157] | PSG | HSAT | Online | 88 | Sensitivity = 98% Specificity = 76% |
| Tanaka et al., 2020 [158] | PSG | WP device | Offline | 774 | - |
| Kay et al., 2021 [159] | PSG | HSAT | Online | 1 | - |
| Bollu et al., 2020 [160] | PSG | nox-T3 sleep monitor and Nomad HSAT | Online | 178 | - |
| Yeh et al., 2020 [161] | PSG | Sleep physicians | Offline | - | - |
| Sterner et al., 2020 [162] | - | WatchPAT | - | - | - |
| Iakoubova et al., 2020 [163] | PSG | Sleep physicians | Online | 900 | - |
| Arguelles et al., 2018 [164] | PSG | Sleep physicians | Online | 60 | Accuracy = 90% |
| Gamaldo et al., 2018 [165] | PSG | HSAT | Online | 147 | - |
| Journal et al., 2019 [166] | PG | Sleep physicians | Online | 1055 | - |

**Table A1.** *Cont.*

| Authors | Signal | Detection Method | Online/Offline | Number of Participants | Detection Performance |
|---|---|---|---|---|---|
| He, Mendez, and Atwood 2020 [167] | PSG | WatchPAT | Online | 295 | - |
| Pinheiro et al., 2020 [168] | PSG | HST | Online | 1013 | Sensitivity = 95.8% Specificity = 94.3% |
| Anderer et al., 2020 [169] | PSG | Deep Learning | Online | 472 | Accuracy = 95%. |
| Zeineddine et al., 2020 [170] | PSG | HSAT | Online | 33 | - |
| F. Facco et al., 2018 [171] | PSG | HST | Online | 34 | Accuracy = 90.5% |
| Zhongming et al., 2021 [172] | PSG | HSAT | Online | 31 | - |
| Carey et al., 2020 [173] | PSG | WPHST | Online | 62 | - |
| Aydin et al., 2020 [174] | PSG | APAP | Online | 43 | - |
| Homan et al., 2021 [175] | SpO$_2$ | HSAT | Online | 558 | Accuracy = 90% |
| Rudock et al., 2019 [176] | PSG | HSAT | Online | - | - |
| Bliznuks et al., 2022 [177] | SpO$_2$ | CPAP | Online | 16 | - |
| Thomas et al., 2021 [178] | PSG | HSAT | Online | 297 | - |
| Kazaglis 2018 [179] | Audio | Noxturnal T3 device | Offline | 2 | - |
| Arguelles et al., 2019 [71] | PSG | HSAT | Online | 11 | Accuracy = 95% |
| Fynn et al., 2020 [180] | PSG | sleep physicians | Online | 246 | - |
| Wenbo et al., 2019 [181] | PSG | ring-type pulse oximeter | Online | 32 | Accuracy = 95.0% |
| Gutiérrez-Tobal et al., 2018 [182] | SpO$_2$ | SAHS | Online | 200 | Sensitivity = 83.8% Specificity = 85.5% |
| Stretch et al., 2020 [183] | PSG | NN approach | Offline | 1329 | 79% |
| Johnson et al., 2018 [184] | - | HSAT | - | - | - |
| Sever et al., 2018 [185] | PSG | Sleep physicians | Online | 1 | - |
| Martinot et al., 2020 [186] | PSG | Machine learning | Online | 192 | Accuracy = 84% |
| Haaland et al., 2018 [187] | PSG | Apnealink | Online | 1021 | - |

**Table A1.** *Cont.*

| Authors | Signal | Detection Method | Online/Offline | Number of Participants | Detection Performance |
|---|---|---|---|---|---|
| Do et al., 2022 [188] | PSG | HSAT | Online | 505 | - |
| Stanchina et al., 2020 [189] | PSG | APAP | Online | 238 | - |
| Perriol et al., 2018 [190] | PSG | CPAP | Offline | 66 | - |
| Krause-Sorio et al., 2021 [191] | HR and SpO$_2$ | Telephone screening | Offline | 5 | - |
| Mahmood et al., 2018 [192] | PSG | HST | Offline | 454 | - |
| Robinson et al., 2018 [193] | PSG | HSAT | Offline | 961 | Sensitivity = 97.1% Specificity = 100% |
| Ferreira 2019 [194] | PSG | CPAP | Online | 191 | - |

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
