# Peer review of "Environmental Benefits of Sleep Apnoea Detection in the Home Environment"

_processes, doi:10.3390/pr10091739_

Round 1
Reviewer 1 Report
This review summaized recently published methods on the detection of sleep apnea. As the thread from SA has been well realized, home monitoring of sleep has shown its importance in the early diagnosis of SA. The review was well written and informative.
Reviewer 2 Report
The authors aimed to establishes the environmental benefits of sleep apnea (SA) detection in the home environment. The manuscript provides a very detailed description of SA and the issues and the authors should be commended for this. The manuscript would benefit having a clear Methods section before presenting results. The literature search was limited to Google Scholar and it would have benefited from using a number of recognised sourced (e.g. PubMed, MEDLINE etc). Further discussion is required surrounding the ethical and privacy concerns as precision medicine and the Internet of Things (IoT) has raised many issues and cannot be understated when investigating personal devices used for data collection and detection.
Some specific comments:
LINES 31-34: Please provide a reference for your comments “A good night sleep can help people to be more productive at work and have a more positive attitude on life. . Sleep deprivation can lead to endocrine problems, raise the risk of amnesia and other diseases, and create inattention, all of which have a negative impact on regular working and living conditions.”
Your regularly mention “environmental impact” in your introduction and background but only define what is meant by this term in lines 179-281 of your Discussion. Please define this term earlier in your manuscript. Please indicate how you would measure “the environmental impact for RM based SA detection.”
Table 1 and Figure 1 provides results from your literature search but the search criteria have not been introduced. Please provide a Methods section.
Pie charts – Please indicate what the figures mean in the pie charts.
Reviewer 3 Report
Thank you for possibility to review the manuscript titled Enviromental benefits of sleep apnoea detection in the home environment.
The topic is of interest and manuscript is well written, however there are many serious flaws.
1) The manuscript is not compliant with the PRISMA guidelines. There is no identification the report as a systematic review.
2) The databases, registers, websites, other sources are not specified. The date when each source was last searched or consulted is not provided
3) There is no hypothesis .
4) Authors should use OSA abbreviation instead of SA. I don’t think they reviewed central apnea and all citated articles are about OSA.
5) 11 12 Controversial. Polysomnography is reliable method of sleep assessment.
6) 33 Firstly , authors should mention cardiovascular problem , which are more prevalent than endocrine ones.
7) 37 The breathing rate is not lowered in sleep apnea. There is reductions in airflow (hypopnea) or absence of airflow ( apnea). In OSA, breathing rate variability can be increased. (Pal a et al. Breathing rate variability in obstructive sleep apnea during wakefulness, JCSM 2022)
8) 47-47 Diagnosis of OSA is based on AHI value and clinical criteria as well.
9) The symptoms are NOT clearly observable in ECG . OSA can be seen in airflow sygnal as respiratory events. It is imposible to diagnose OSA using ECG only. OSA can be suspected because of ECG variability.
10) PSG can be recorded more than one night. The adaptation night is commonly conducted.
11) 122-124 OSA is mainly detected via Airflow signal. The termistor or canula are used.
12) 127 The ECG sensitivity and specificity is too low to diagnose OSA.
13) 159 Incorrect. Normal arterial blood oxygen saturation levels in humans are 95–100 percent.
14) 173 Most of patient have no trouble sleeping in sleep lab. Provide citation, if there is such a study.
15) Table 1 and Figures -Abbreviations should be expanded.
16) 235 In result section there is 114 articles , however in flowchart there is 113. One is missing?
Here I stop.
Round 2
Reviewer 3 Report
The authors have made few corrections. The manuscript looks superficially better ,however there are still shorcomings and controversial opinions.